# Underlying dyslipidemia postpartum in women with a recent GDM pregnancy who develop type 2 diabetes

Mi Lai[1], Dana Al Rijjal[1], Hannes L Röst[2]*, Feihan F Dai[1]*, Erica P Gunderson[3]*, Michael B Wheeler[1,4]*

[1]Department of Physiology, Faculty of Medicine, University of Toronto, Ontario, Canada; [2]Donnelly Centre for Cellular & Biomolecular Research, University of Toronto, Ontario, Canada; [3]Kaiser Permanente Northern California, Division of Research, Oakland, United States; [4]Advanced Diagnostics, Metabolism, Toronto General Research Institute, Ontario, Canada

**Abstract** Approximately, 35% of women with Gestational Diabetes (GDM) progress to Type 2 Diabetes (T2D) within 10 years. However, links between GDM and T2D are not well understood. We used a well-characterised GDM prospective cohort of 1035 women following up to 8 years postpartum. Lipidomics profiling covering >1000 lipids was performed on fasting plasma samples from participants 6–9 week postpartum (171 incident T2D vs. 179 controls). We discovered 311 lipids positively and 70 lipids negatively associated with T2D risk. The upregulation of glycerolipid metabolism involving triacylglycerol and diacylglycerol biosynthesis suggested activated lipid storage before diabetes onset. In contrast, decreased sphingomyelines, hexosylceramide and lactosylceramide indicated impaired sphingolipid metabolism. Additionally, a lipid signature was identified to effectively predict future diabetes risk. These findings demonstrate an underlying dyslipidemia during the early postpartum in those GDM women who progress to T2D and suggest endogenous lipogenesis may be a driving force for future diabetes onset.

*For correspondence:
hannes.rost@utoronto.ca (HLR);
f.dai@utoronto.ca (FFD);
Erica.Gunderson@kp.org (EPG);
michael.wheeler@utoronto.ca
(MBW)

## Introduction

Gestational diabetes mellitus (GDM) develops during pregnancy, affecting 1–14% of all pregnancies depending on diagnostic criteria and the population characteristics (*Chen et al., 2018*; *Melchior et al., 2017*). The majority of women with a history of GDM were not known to have overt diabetes before pregnancy and return to non-diabetes post-delivery. However, women with a history of GDM are ~7 times more likely to develop type 2 diabetes (T2D) during the child-bearing years compared to women who had no previous GDM (*Chen et al., 2018*; *Bellamy et al., 2009*; *Gunderson et al., 2007*). In fact, it is estimated that 35–50% of women with GDM may progress to T2D within 10 years after delivery (*Bellamy et al., 2009*; *Tobias, 2018*). Within 15 to 25 years, the lifetime maternal risk for overt diabetes is estimated to reach >50% (*American Diabetes Association, 2019*; *Kim et al., 2002*). Therefore, it is critical to uncover the underlying metabolic changes and understand the distinctive pathophysiology in T2D progression/development following GDM.

In the past decade, omics-based approaches have been used to discover novel metabolic fluctuations in humans, providing insight into pathophysiology of disease and identifying biomarkers of future disease including diabetes (*Sas et al., 2015*; *Khan et al., 2019*; *Allalou et al., 2016*). In particular, lipidomics has emerged as a more specialized omics platform that enables the measurement of a wide spectrum of lipid species. This approach has greatly expanded our understanding of the complexity of lipid dysregulation in metabolic diseases. Recently, an increasing number of lipidomics studies have aimed to link lipid dysregulation to diabetes pathology (*Meikle et al., 2013*; *Lu et al.,*

*2018*; *Rhee et al., 2011*; *Lu et al., 2019*; *Meikle et al., 2014*; *Alshehry et al., 2016*; *Lu et al., 2016*; *Suvitaival et al., 2018*; *Razquin et al., 2018*). In the Framingham Heart Study cohort, more than 100 lipid analytes were measured and a group of triacylglycerols (low total carbon number and carbon double bonds) were found to be associated with increased risk of T2D (*Rhee et al., 2011*). In the PREDIMED trial, 207 plasma lipids were measured in which lysophosphatidylcholines (LPCs), phosphatidylcholine-plasmalogens (PC-PLs), sphingomyelins (SMs), and cholesteryl esters (CEs) were found to be inversely associated with T2D risk while triacylglycerols (TAGs), diacylglycerol (DAGs) and phosphatidylethanolamine (PEs) were positively associated with T2D risk (*Razquin et al., 2018*). A total of 277 plasma lipids were analyzed using a lipidomics approach in Finnish males in which five lipids were selected to predict progression to Type 2 diabetes (T2D) (*Suvitaival et al., 2018*). In this cohort, higher levels of specific TAGs and diacyl- phospholipids and lower levels of alkylacyl-phosphatidylcholines were also observed in those who progressed to T2D (*Suvitaival et al., 2018*). In a very recent lipidomics study of a Chinese cohort, 250 lipids were tested and 38 significantly associated with T2D risk, including TAGs, LPCs, PCs, polyunsaturated fatty acid (PUFA)–plasmalogen phosphatidylethanolamines (PUFA-PEps), and CEs (*Lu et al., 2019*). A lipid panel including six lipids significantly improved T2D prediction compared to that achieved by conventional risk factors (*Lu et al., 2019*). In all of these studies, the positive association of TAG/DAG and T2D risk was consistently reported. However, a convergence on other specific lipids were not evident. This could be due to the differences in study design, cohort background and methodology including, importantly, limitations in coverage - expressed lipids in each study were not consistent.

Lipidomics has also been performed in GDM cohorts, including the measurement of 181 lipids in serum samples obtained from GDM women in their early second trimester. Four lipid biomarkers (TG(51:1), TG(48:1), PC(32:1), and PCae(40:4)) were identified for GDM prediction with a moderate accuracy 71% (*Lu et al., 2016*). Another lipidomic study measuring ~300 lipid species in blood samples from 104 women with recent GDM at 12 weeks post-delivery, of whom 21 cases later developed T2D, showed 84% accuracy in T2D prediction based on three lipids [i.e., PE(P-36:2), PS38:4, CE20:4] in combination with six other risk factors (i.e., age, BMI, prenatal fasting glucose, postpartum fasting glucose, total triglycerides, and total cholesterol) (*Lappas et al., 2015*). Our team identified seven lipids from early postpartum blood samples to predict later incident T2D with an AUC of 0.92 in a relatively small subset of women with recent GDM in our large prospective cohort (55 matched pairs of incident cases controls) (*Khan et al., 2019*). To date however, no consensus has been achieved in terms of lipidomic dysregulation in GDM progression to T2D, likely due to limitation in the coverage of lipidome, cohort size, clinical data including diagnosis and follow-up years. Lipidomic changes within a large prospective cohort of women with GDM followed from the early postpartum period have not been evaluated. A comprehensive evaluation of lipidomic changes in relation to progression to T2D could elucidate the pathogenesis of transition from GDM to T2D, and thereby improve our understanding of the clinical targets for therapeutic interventions.

In the present study, lipidomics of 1008 lipid species from 15 lipid classes and 296 fatty acids was measured in a well-characterised prospective cohort with recent GDM pregnancy and no diabetes, followed from 6 to 9 weeks post-delivery (baseline), retested with OGTTs for 2 years and followed via clinical laboratory testing and diagnoses up to 8 years later. Our aims were to systematically investigate lipidomic dysregulation in the transition from no diabetes to incident T2D following a GDM pregnancy and uncover lipid markers that may facilitate the early prediction of T2D incidence with clinical risk factors.

## Results

### Clinical characterization of the participants at baseline

The SWIFT cohort enrolled a total of 1035 women diagnosed with GDM. Of these, 1010 did not have T2D at 6–9 weeks postpartum (baseline) and 989 had follow up testing for glucose tolerance up to 8 years post-baseline. Fasting blood samples were collected at baseline. During the follow-up period, 197 women had developed incident T2D and 791 did not (*Figure 1*). The total years of follow-up were similar between incident T2D and control groups. All research participants underwent 2 hr 75 g OGTTs and other assessments at baseline and thereafter annually for 2 years and subsequent medical diagnoses of diabetes was retrieved from electronic medical records for 8 years post-

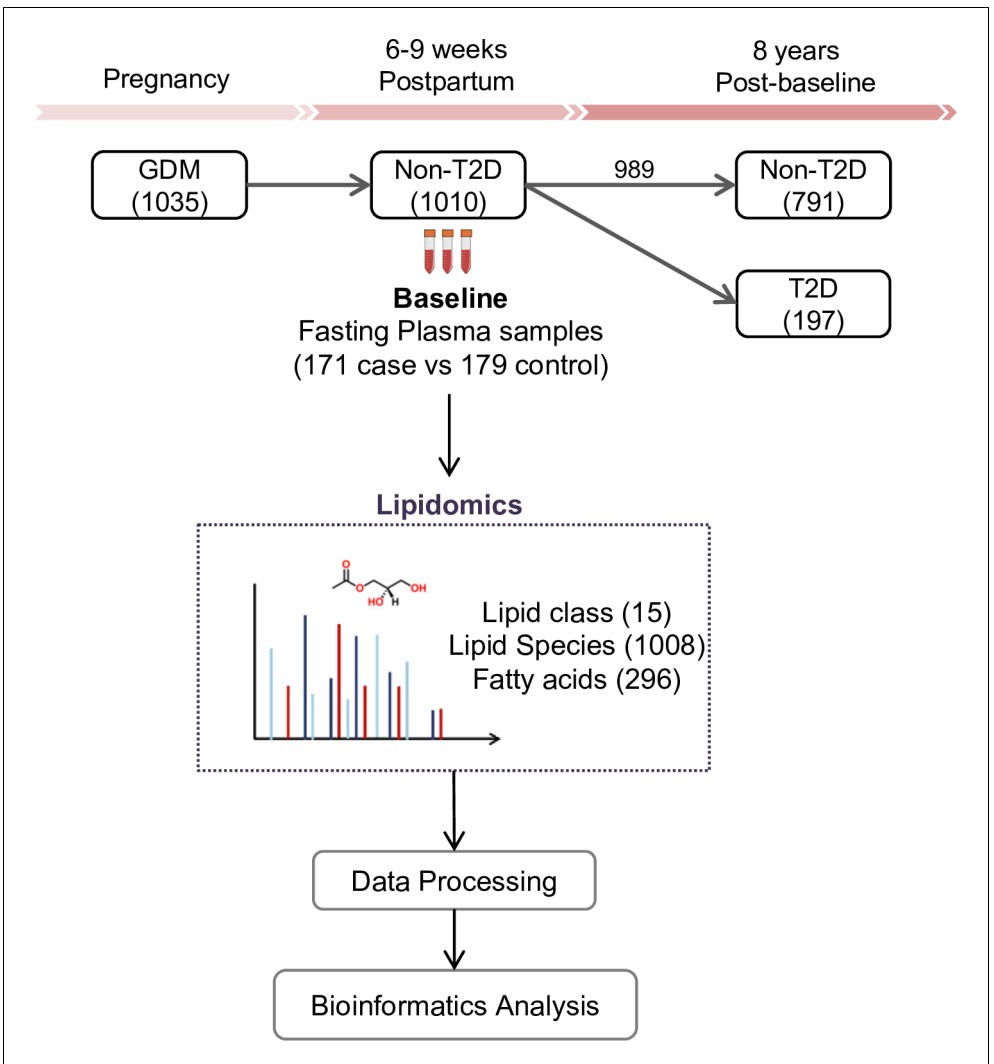

**Figure 1.** SWIFT cohort and study design. In the SWIFT cohort, 1035 women diagnosed with GDM in 2008–2011 were enrolled at 6–9 weeks postpartum (baseline). 1010 of the 1035 participants were confirmed via 2 hr 75 g OGTT without diabetes at baseline. Up to 8 years' post-baseline, a total of 197 (19.5%) women developed T2D. At baseline, samples of 171 available cases with 179 controls were measured using lipidomics. A total of 1008 lipid species from 15 lipid classes and 296 fatty acids were assessed in the plasma samples of all participants.

baseline. In our current study, 171 women with incident T2D cases had available plasma samples at baseline, and 179 controls who did not develop T2D in 8 years' follow-up (350 participants in total) were profiled for lipidomics. A total of 1008 lipid species from 15 lipid classes as well as 296 fatty acids were assessed in the plasma samples of all participants (*Figure 1*). Socio-demographic and clinical parameters of the 350 participants at baseline are summarized in *Table 1*. There was no significant difference in age, race, parity, pre-pregnancy BMI, family history of diabetes, postpartum BMI, total cholesterol, LDL-C, HOMA-B, smoker, dietary glycemic index, dietary intake and physical activity score. Compared to the control group, a higher percentage of participants who developed T2D later on had been treated with insulin or oral medications during pregnancy (p<0.001). Prenatal 3 hr 100 g OGTT (sum of the 4 z-scores for glucose values; fasting, 1 hr, 2 hr and 3 hr post-load, p<0.001) for the incident T2D case group were higher than the control group. At 6–9 weeks postpartum, compared to controls, women in the incident T2D group had higher mean FPG (p<0.001), 2hPG (p<0.001), fasting insulin (p=0.001), 2 hr insulin (p<0.001), fasting TAG (p=0.003), median HOMA-IR (p<0.001) and hypertension (p=0.04), but lower mean fasting HDL-C (p=0.017).

**Table 1.** Prenatal and study baseline (6–9 weeks postpartum) characteristics of women with gestational diabetes mellitus in the SWIFT cohort (n = 350).

| Prenatal characteristics | Case Diabetes at follow up (N = 171) | Control No Diabetes at follow up (N = 179) | p-value |
|---|---|---|---|
| Age (years), Mean (SD) | 33.3 (5.2) | 33.0 (4.5) | 0.63 |
| Race/ethnicity, n (%) | | | 0.72 |
| White | 31 (18.1) | 27 (15.1) | |
| Asian | 51 (29.8) | 55 (30.7) | |
| Black | 21 (12.3) | 16 (8.9) | |
| Hispanic | 66 (38.6) | 79 (44.1) | |
| Other | 2 (1.2) | 2 (1.1) | |
| Parity, n (%) | | | 0.80 |
| Primiparous (one birth) | 56 (32.7) | 54 (30.2) | |
| Biparous (two births) | 62 (36.3) | 64 (35.8) | |
| Multiparous (>2 births) | 53 (31.0) | 61 (34.1) | |
| GDM treatment, n (%) | | | <0.001 |
| Diet only | 74 (43.3) | 128 (71.5) | |
| Oral medications | 79 (46.2) | 47 (26.3) | |
| Insulin | 18 (10.5) | 4 (2.3) | |
| Pre-pregnancy BMI (kg/m$^2$), Mean (SD) | 33.6 (8.2) | 32.3 (6.9) | 0.10 |
| Sum of Prenatal 3 hr 100 g OGTT glucose z-scores, Mean (SD) | 1.4 (3.1) | −0.2 (2.5) | <0.001 |
| Family history of diabetes, n (%) | 101 (59.1) | 89 (52.0) | 0.08 |
| Baseline characteristics at 6–9 weeks Postpartum | | | |
| BMI (kg/m$^2$), Mean (SD) | 33.5 (7.4) | 32.4 (6.3) | 0.18 |
| Fasting plasma glucose (FPG), mg/dl, Mean (SD) | 101.5 (10.4) | 94.3 (7.7) | <0.001 |
| 2 hr Post-load plasma glucose (75 g OGTT), mg/dl, Mean (SD) | 131.0 (29.5) | 109.8 (27.4) | <0.001 |
| Fasting insulin, μU/ml, Median (IQR) | 26.5 (20.7) | 22.1 (17.4) | 0.001 |
| 2 hr insulin, μU/ml, Median (IQR) | 111.5 (85.7) | 83.3 (73.6) | <0.001 |
| Fasting plasma Triglycerides, mg/dl, Median (IQR) | 119.0 (103.0) | 94.0 (72.0) | 0.003 |
| Fasting plasma HDL-C, mg/dl, Mean (SD) | 49.0 (16.0) | 52.0 (19.0) | 0.017 |
| Fasting plasma Total Cholesterol, Mean (SD) | 199.4 (34.5) | 203.5 (35.5) | 0.27 |
| Fasting plasma LDL-C, Mean (SD) | 121.0 (31.1) | 126.4 (31.2) | 0.10 |
| HOMA-IR, Median (IQR) | 6.8 (5.6) | 5.0 (4.3) | <0.001 |
| HOMA-B, Median (IQR) | 268.1 (192.1) | 256.0 (176.2) | 0.61 |
| Hypertension, n (%) | 14 (8.2) | 5 (2.8) | 0.04 |
| Smoker, n (%) | 5 (2.9) | 4 (2.2) | 0.68 |
| Dietary glycemic index, Mean (SD) | 242.5 (106.7) | 246.5 (112.5) | 0.73 |
| Dietary Intake, Percentage of Kcal as animal fat, Mean SD | 27.0 (7.7) | 25.6 (8.6) | 0.10 |
| Physical activity score, met-hrs per week, Mean (SD) | 50.7 (23.4) | 47.4 (20.6) | 0.16 |

Variables obtained from the SWIFT Study that administered the research 2 hr 75 g OGTTs and other assessments at in-person research visits (baseline). Participants did not have diabetes at study baseline and underwent annual 2 hr 75 g OGTTs at baseline and annually for two years, and thereafter evaluated for diabetes onset from electronic medical records. P-values are for incident diabetes case versus no diabetes controls at follow-up.

## Lipids associated with future T2D risk

Lipid biosynthesis and metabolism have been implicated in the development and progression of T2D. However, in previous studies, it has been an understudied component of metabolomics

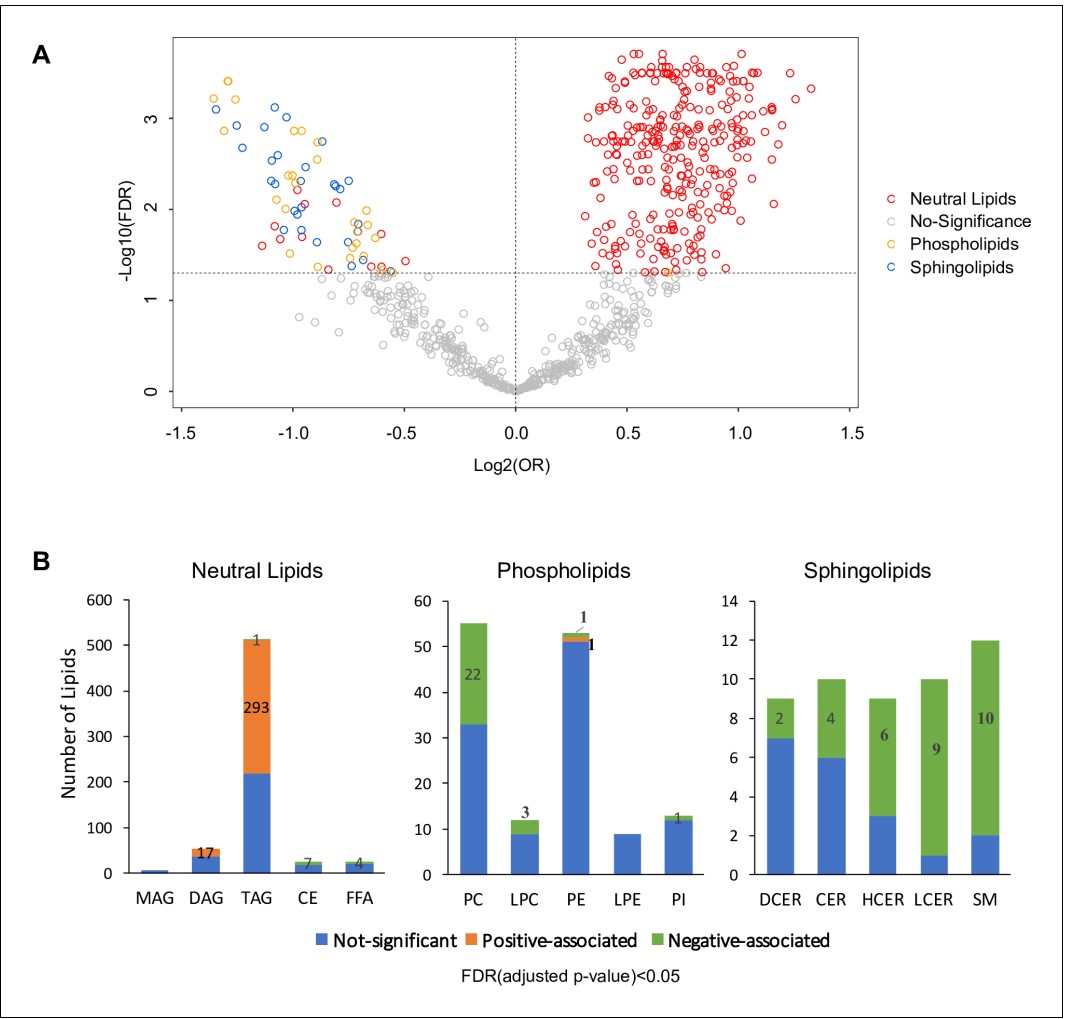

**Figure 2.** Overview of T2D associated lipids. (**A**) Volcano plot showed -log10(FDR) against log2(OR) of 816 lipid species in the association with T2D risk. Grey circles were denoted as no significant association with T2D risk. Of those that are significantly associated, red circles denote as neutral lipids, orange as phospholipids, blue as sphingolipids. (**B**) Number of T2D positive-, negative- and non- associated lipids in each lipid class were shown. Orange, green and blue bars denote positive-, negative- and non- associated lipids respectively. Significance was indicated by FDR < 0.05.

The online version of this article includes the following source data and figure supplement(s) for figure 2:

**Source data 1.** Odds ratio, 95% CI and FDR values of all lipids. Lipids with FDR < 0.05 were highlighted.
**Figure supplement 1.** Supervised PCA indicated partial separability of lipid profiles between case and control groups.
**Figure supplement 2.** Odds ratio and 95% CI of 311 lipids associated with T2D risk (FDR < 0.05).

profiling in the GDM transition to T2D. Thus, we have launched a broad spectrum lipidomics analysis, screening lipid metabolites and providing a comprehensive linkage of lipid metabolism to T2D. With a total of 1008 lipid species, we excluded lipids with >5% missing values among subjects, allowing only robust lipids (816 species) to be included in further analysis. Supervised PCA indicated partial separability of lipid profiles between case and control groups (*Figure 2—figure supplement 1*). By applying multiple logistic regression analysis, we assessed the association of lipids with future diabetes risk after adjusting for age, race and BMI. Of the 816 lipid species, 311 were positively and 70 were negatively associated with T2D risk (*Figure 2A*, *Figure 2—figure supplement 2*, FDR < 0.05). Of the 311 lipids positively associated with risk, 293 were from TAG class while 17 from DAG class and one from PE class (*Figure 2A–B*). Of the 70 lipids negatively associated with T2D, 31

were from SM class, 27 from PC class, seven from CE class, four from FFA class and one from TAG class (*Figure 2A–B*).

Most notably, 57.2% of all TAG species measured (293 out of 512 TAG) were significantly positively associated with T2D risk (*Figure 2B*). Plasma TAG, a transporter of dietary fats, increased, suggesting an overload of lipids in circulation before T2D onset. Additionally, 17 out of 54 DAGs, intermediates of TAG synthesis, were upregulated, further suggesting TAG biosynthesis was abnormally active (*Figure 2B*). In contrast, 40% (22 out of 55) measured PC and 25% (3 out of 12) measured LPC were negatively associated with T2D risk (*Figure 2B*). Similarly, 62% measured sphingolipids (31 out of 50) were inversely associated with T2D risk, particularly in classes of HCER (6 out of 9), LCER (9 out of 10) and sphingomyelins (10 out of 12) (*Figure 2B*). These findings suggested an inverse association of phospholipids and sphingolipids and increased risk of T2D.

More strictly, by using a cut-off of FDR < 0.001, we demonstrated 107 lipids were significantly associated with T2D progression (*Figure 3A*). In this panel, 97 TAGs spanning carbon atom numbers from 42 to 56 with double bonds from 0 to 8, along with one saturated DAG(16:0/16:0) were consistently associated with increased diabetes risk. One monounsaturated PC(17:0/18:1) and three polyunsaturated PC(17:0/18:2), PC(18:1/20:4), PC(18:2/16:1) were inversely associated with future diabetes risk. Similarly, SM (18:1), SM(20:1), SM(24:1), HCER(24:1), and LCER(16:0) from the sphingolipid class were negatively associated with diabetes risk. Correlations between the 107 incident T2D associated lipids and conventional clinical parameters (BMI, FPG, 2hPG, fasting insulin HOMA-IR and HOMA-B) were assessed (*Figure 3B*). TAGs and DAG demonstrated a weak to moderate positive correlation with fasting insulin and HOMA-IR while sphingolipids and phospholipids were shown to have a weak negative correlation (*Figure 3B*). In contrast, those 107 lipids showed little correlation with 2hPG, age and BMI (*Figure 3B*).

## Association between diabetes risk and lipid biochemical configuration

Lipidomics profiling provided a comprehensive coverage of plasma lipids for us to gain insight into the associations of lipid species biochemical structure (i.e. chain length, numbers of carbon atoms, double bonds) with diabetes risk. Among all the TAGs detected (carbon atoms from 36 to 60), those significantly associated with diabetes risk contained between 40–56 carbon atoms and 0–8 double bonds. Within those TAGs containing 40–56 carbon atoms, T2D risk increased in step with the number of carbon atoms (except carbon atom 55). TAGs most significantly associated with T2D risk were clustered in the range of carbon atoms 50–54 and double bond 0–4, particularly with even carbon atoms 52 and 54 (*Figure 4A*). DAGs with an even number of carbon atoms 30, 32, 34, 36 but not odd numbers were associated with diabetes risk more prominently. There was no clear pattern of association with incident T2D by numbers of carbon atoms or double bonds in other lipid classes (*Figure 4A*). From the perspective of specific fatty acid chains in lipids, a relationship between diabetes risk and fatty acid composition was revealed. For total fatty acids, three SFAs (FA12:0, FA14:0 and FA16:0) as well as a PUFA (FA18:3) were positively associated with T2D risk and two very long chain MUFAs (FA24:1, FA26:1) were negatively associated with T2D risk (*Figure 4B*). Considering lipid classes, positively associated fatty acids were mainly from DAGs and TAGs including long chain SFAs (C12-C20), MUFA (C14 and C16) and PUFA (C20 and C22) (*Figure 4B*). In contrast, in PC and LPC classes, odd chain fatty acids (C15 and 17) were negatively associated with T2D risk. Interestingly, in the sphingolipid class, only even chain saturated and MUFAs were negatively associated with T2D risk (*Figure 4B*).

## Metabolic pathways associated with future diabetes

To identify metabolic pathways associated with future diabetes, 381 lipids with significant association with diabetes risk (FDR < 0.05) (*Figure 2—figure supplement 2*) were subjected to Kyoto Encyclopedia of Genes and Genomes (KEGG) pathway analysis. Glycerolipid metabolism, which involves TAG and DAG biosynthesis, was significantly up-regulated (p=0.01). In contrast, sphingolipid (p=2.11E-05), linoleic acid (p=0.016) and alpha linoleic acid (p=0.041) metabolism were found to be significantly down-regulated (*Figure 5A*). Specifically, in the glycerolipid biosynthesis pathway, the TAG class was increased with strong significance (p=0.003), suggesting an induced process of lipid storage (*Figure 5B*). While as a whole the phospholipid metabolism pathway was not significantly altered, the PC class of lipids was significantly reduced (p=0.015) along with a modest decrease in

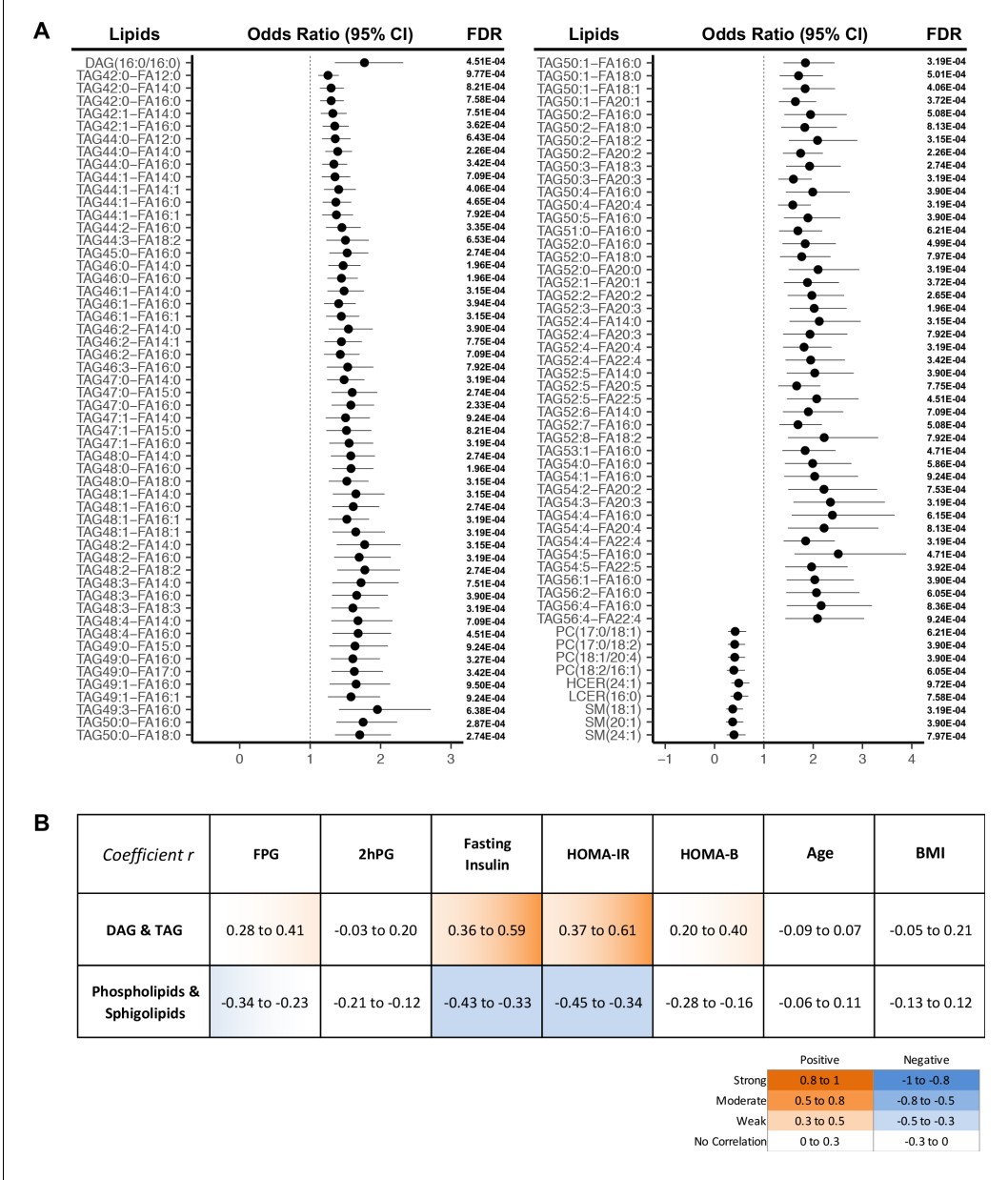

**Figure 3.** Lipids strongly associated with risk of incident T2D. (**A**) Odds ratio and 95% CI of 107 lipids strongly associated with T2D risk (FDR < 0.001) were indicated. The multivariate logistic regression model was adjusted for race, age and BMI. (**B**) Correlation between 107 T2D-risk associated lipids and conventional clinical parameters was indicated by correlation coefficient (r). Orange color indicates positive correlation while blue denotes negative correlation.

The online version of this article includes the following source data for figure 3:

**Source data 1.** Odds ratio, 95% CI and FDR values of all lipids. Lipids with FDR < 0.001 were highlighted.

**Source data 2.** Correlation r values of T2D risk associated lipids with clinical parameters.

the downstream LPC class (p<0.2), suggesting the potential inhibition of pathway from DAG to PC class. In sphingolipids metabolism, the central metabolite ceramide, which is a precursor for complex sphingolipids, was marginally down-regulated (p<0.2). However, classes of SM (p=0.002), HCER (p=0.006) and LCER (p=0.0005), which are downstream of sphingolipid metabolism were highly

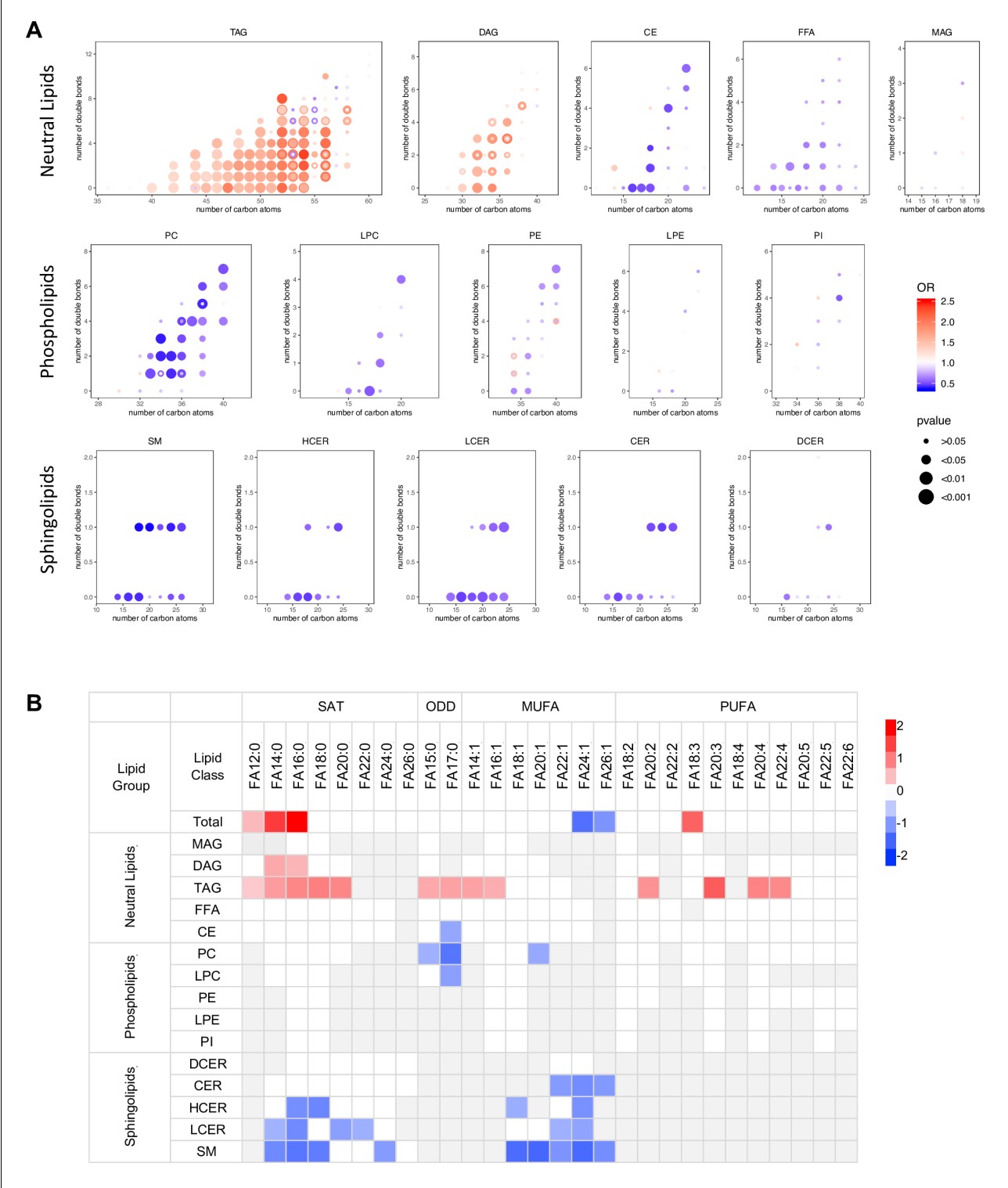

**Figure 4.** Association between diabetes risk and lipid structure. (**A**) Relationship between diabetes risk and total number of carbon atoms and double bonds in lipid species. Odds ratios were represented with dots, color denoting odds ratio value, dot size denoting significance by FDR value. (**B**) Relationship between diabetes risk and fatty acid composition in lipids. Red and blue color denotes log2(odds ratio) with significance (FDR < 0.05), white denotes values with no significance, grey denotes fatty acids not detected.

The online version of this article includes the following source data for figure 4:

**Source data 1.** Odds ratio values, FDR values, numbers of carbon atoms and double bonds in all lipid species.

**Source data 2.** Relationship between diabetes risk and fatty acid composition in lipids.

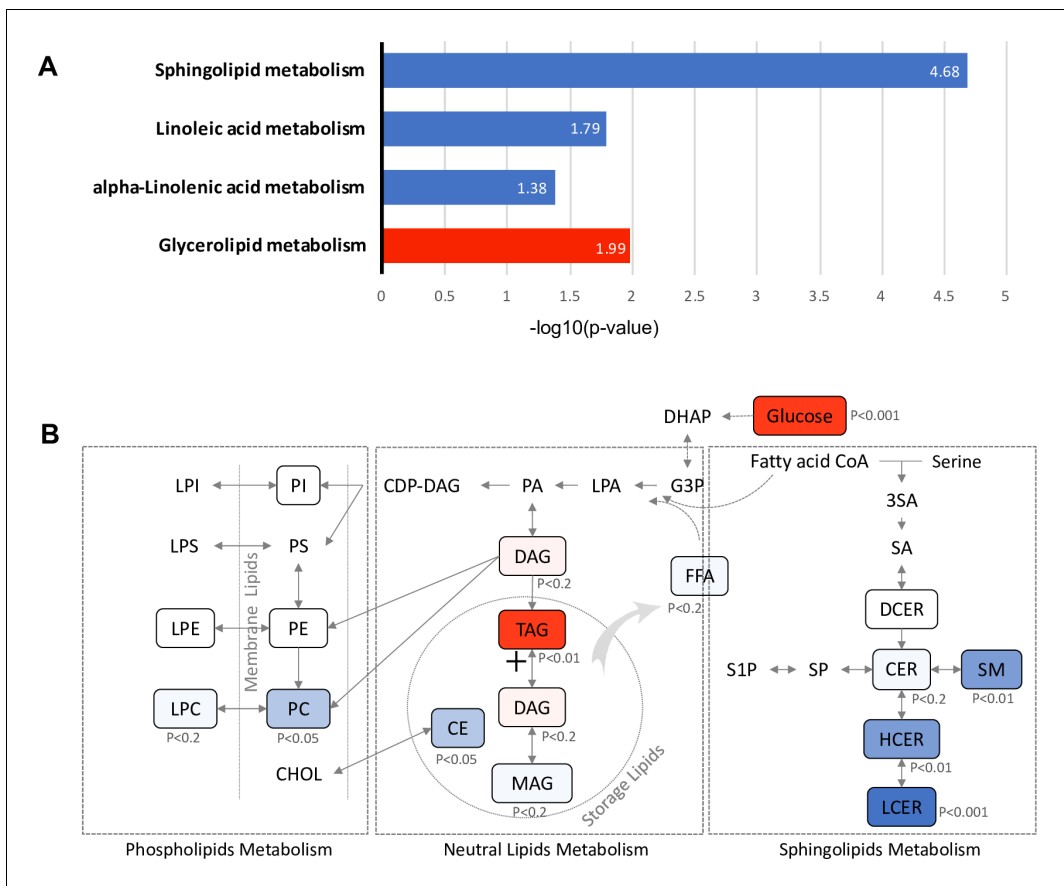

**Figure 5.** Pathways associated with future T2D at baseline. (**A**) Significantly regulated biological pathways associated with future diabetes onset analyzed by Kyoto Encyclopedia of Genes and Genomes (KEGG). Blue denotes the down-regulated pathways and red denotes the up-regulated pathway. (**B**) The altered lipid classes in an integrated lipid metabolism pathway. Red denotes positive association whereas blue denotes negative association with significance of p-value indicated.

reduced, suggesting the inhibition in the process of deriving complex sphingolipids from ceramide (*Figure 5B*).

## Selective lipids can predict future diabetes and complement clinical diagnostics

The 107 lipids are the most significantly associated with future diabetes (odds ratio FDR cut-off <0.001) (*Figure 3A*). It is intuitive that some may actually have predictive properties, and this was tested. By using stepwise logistic regression modelling, we identified a panel of 11 lipids (10 TAGs and 1 PC) with excellent ability to predict future diabetes in the cohort examined (*Figure 6A*). With these lipids alone, we achieved the prediction ability as AUC of 0.739 (*Figure 6B*). The classical clinic predictive parameter FPG showed the prediction power of AUC 0.703 which was improved to AUC 0.795 by adding lipids (*Figure 6B*). The clinic predictive parameter 2hPG showed the prediction power of AUC 0.704 which was improved to AUC 0.809 by adding lipids (*Figure 6B*). The combination of two clinical parameters 2hPG and FPG can achieve an AUC 0.775. Importantly, combining the 11 lipid panel outcomes with FPG and 2hPG, the discriminative power was significantly improved to AUC 0.842 (*Figure 6B*). This demonstrates that the circulating levels of specific lipids can in part be used to assess future diabetes risk and when applied, can improve diabetes prediction, especially when combined with routine clinical parameters (2hPG and FPG) during the early postpartum period.

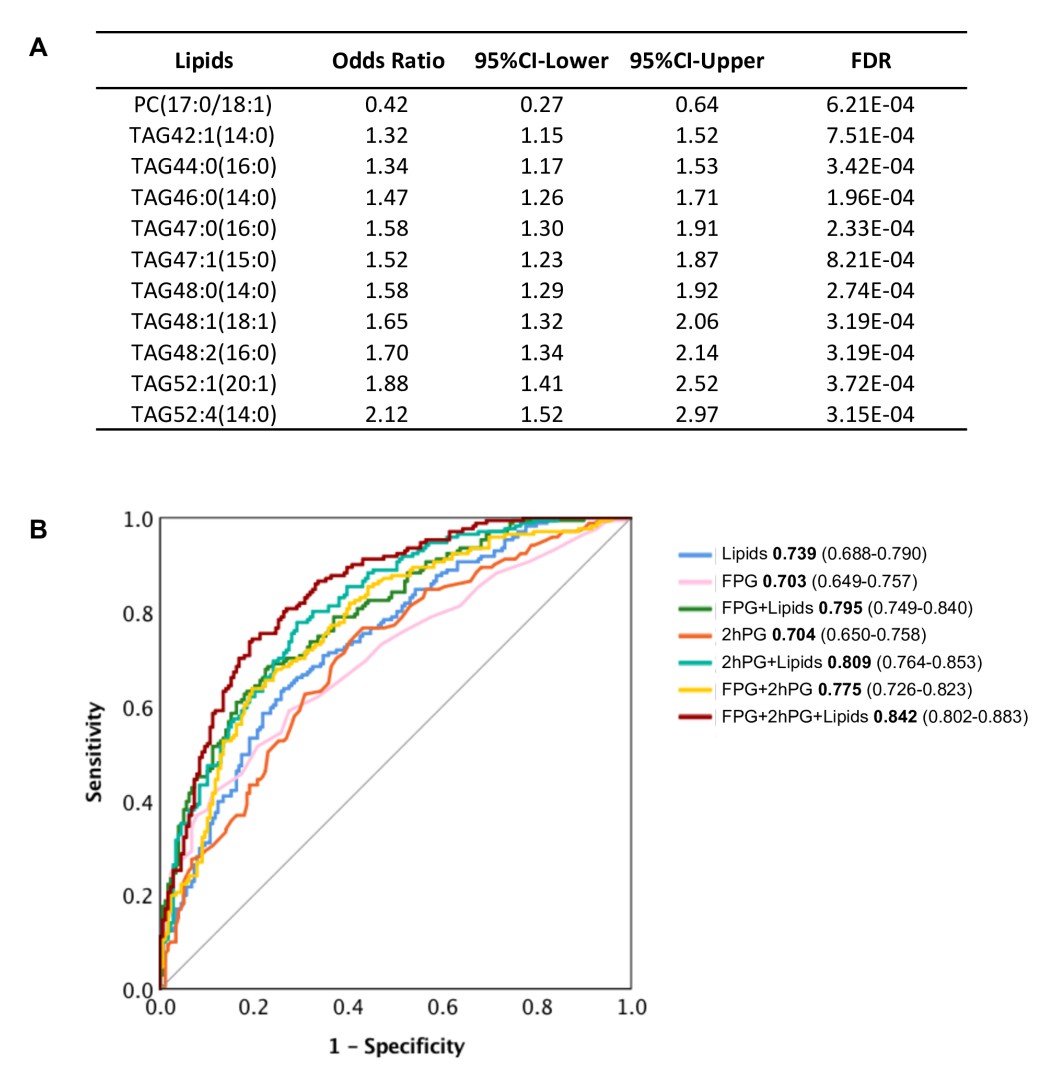

**Figure 6.** Selected lipid signature predicting future T2D. (**A**) Top 11 lipids with the best predictive performance were selected for building a model to predict future T2D. Their odds ratio and 95% CI of T2D association were shown. (**B**) Predictive performance of logistic regression model was demonstrated as ROC curve. The area under the curve and 95% CI in each model were shown.

The online version of this article includes the following source data for figure 6:

**Source data 1.** Predictive performance of logistic regression model.

## Discussion

In the present study, lipidomic profiling was used to assess the lipid changes at early post-partum (6 to 9 weeks) in a well-characterized, racially and ethnically diverse prospective cohort of postpartum women with recent GDM. A lipid signature associated with future diabetes risk was uncovered which contributes new knowledge to understanding the aetiology of diabetes in women associated with GDM. Importantly, our data indicate that women with recent GDM who later develop new onset T2D have clear differences in their lipidome compared to controls after delivery. This clearly shows they already exhibit lipid dysregulation in the early post-partum period.

Among the 311 lipids positively associated with progression to T2D, we found 293 belonging to TAG classes. This is equivalent to an impressive 57.2% of all measured TAGs (293 out of 512) (*Figure 2B*). In addition, among the lipids associated with the most significant T2D risk, 91% of them were TAGs (97 out of 107) (*Figure 3A*). This finding fits our clinical measurements showing elevated TAGs in T2D incident cases (*Table 1*) and is consistent with other studies (*Khan et al., 2019*;

*Meikle et al., 2013*; *Rhee et al., 2011*; *Lu et al., 2019*; *Meikle et al., 2014*; *Alshehry et al., 2016*; *Suvitaival et al., 2018*; *Razquin et al., 2018*; *Lappas et al., 2015*). TAGs, belonging to neutral lipids, are the energy storage in adipocytes and are an efficient energy source for muscle. In plasma, TAGs enable the bidirectional flow of fat from adipose tissue storage and blood glucose from the liver. Therefore, it is not surprising that TAGs outweigh other lipids as the dominant lipid species in terms of reflecting the changes of lipid metabolism in the body. The source of TAGs could be from food intake or endogenous TAG biosynthesis, such as lipogenesis. Our KEGG analysis demonstrated that the glycerolipid metabolism pathway was upregulated, suggesting the accumulation of TAGs could be attributed to the up-regulation of TAG biosynthesis (*Figure 5*). It was reported high sugar could stimulate de novo lipogenesis in liver thereby increasing serum TAG level (*Schwarz et al., 2003*). This process could be activated directly through transcriptional factor carbohydrate responsive element binding protein (ChREBP) to promote expression of lipogenic enzymes. Alternatively, lipogenesis could also be regulated by insulin through sterol regulatory element binding protein-1 (SREBP1). The elevated level of plasma hexose and insulin in those incident T2D cases at baseline could be associated with the enhanced endogenous lipogenesis.

In contrast, classes of glycerophospholipids (PC and LPC classes) are inversely associated with T2D risk (*Figure 2B*). Glycerophospholipids (through DAG) and TAGs share the same precursor glycerol-3-phosphate. Therefore, the downward trend in glycerophospholipids could be linked to the up-regulation of TAG biosynthesis. In addition to the phospholipids, an impressive 62% of measured sphingolipids (31 out of 50 tested) were inversely associated with T2D risk (*Figure 2B*). Particularly SM(18:1), SM(20:1), SM(24:1), HCER(24:1), and LCER(16:0) were among the lipids with the most significant risk associated with diabetes (*Figure 3A*). KEGG analysis revealed that sphingolipid metabolism was the most significantly down-regulated (p=2.11E-05), further supporting the inverse association between sphingolipids and diabetes risk. So far, the relationship between sphingolipids and T2D risk has not been unequivocally ascertained. Several cross-sectional clinical studies have shown that CERs (upstream node of the sphingolipids pathway) are elevated in obese subjects with T2D (*Meikle et al., 2013*; *Lemaitre et al., 2018*; *Lopez et al., 2013*; *Haus et al., 2009*). We and others, however, have previously shown a negative association of SMs (downstream node of the whole pathway) with diabetes risk (*Khan et al., 2019*; *Allalou et al., 2016*; *Razquin et al., 2018*; *Fall et al., 2016*; *Floegel et al., 2013*). Further biological testing in humans and models of diabetes risk are required to validate the association between sphingolipids and diabetes onset.

Glycerophospholipids (through DAG) and TAGs share the same precursor glycerol-3-phosphate (G3P). The higher G3P induced by higher plasma glucose levels could shift the acyl-CoA to lipogenesis from sphingolipids and phospholipids pathways. Therefore, in those incident T2D cases, the downward trend in glycerophospholipids and sphingolipids could be associated with the up-regulation of TAG biosynthesis. In normal physiological conditions, de novo lipogenesis mainly occurs in the liver and adipose tissue and is a minor contributor to serum TAG homeostasis. However, an up-regulated lipogenesis could break the balance causing lipidemia. In addition, down-regulation of glycerophospholipids and sphingolipids biosynthesis impairs the integrity of cell membrane structure, which might contribute to insulin resistance. Although higher glucose level could correlate with higher TAG, TAG is not simply an indirect measure of glucose. Instead, increased TAG along with decreased phospholipids and sphingolipids could be an early sign of up-regulated endogenous de novo lipogenesis, a driving force of T2D.

Investigating the composition of the fatty acids in the lipids showed long chain SFA myristic acid (C14:0) and palmitic acid (C16:0) were positively associated with T2D risk. Previously, palmitic acids were reported to cause pancreatic beta cell dysfunction and were shown to be associated with diabetes (*Oh et al., 2018*; *Nemecz et al., 2018*). A previous study on a large prospective cohort EPIC-InterAct case suggested that even-chain SFA in phospholipids were positively associated with diabetes risk while odd-chain SFA had a negative association (*Forouhi et al., 2014*). Similarly, we detected odd-chain SFA from phospholipids were negatively associated with T2D risk. However, the association between even-chain SFAs and T2D risk was more complicated depending on the lipid classes from which they were derived. Even-chain SFAs from glycerol lipids (TAGs and DAGs) were positively associated with T2D risk while those from sphingolipids had a negative association. No significant association to T2D risk was detected in even-chain SFAs from phospholipids (*Figure 4B*). Odd-chain SFAs (C15:0 and C17:0) are mainly exogenously derived from dairy fat intake (*Smedman et al., 1999*; *Wolk et al., 1998*; *Hodson et al., 2008*). In contrast, even-chain

SFAs are from an endogenous source, such as increased lipolysis from adipose tissue or de-novo lipogenesis from excess carbohydrates (*Hodson et al., 2008*; *Siler et al., 1999*; *Hudgins et al., 1998*; *King et al., 2006*; *Hudgins et al., 1996*).

In addition to the carbon numbers of fatty acids, we also showed the association between the degree of fatty acid unsaturation (number of double bonds) and diabetes. MUFAs, particularly those from sphingolipids, were negatively associated with T2D risk; however,PUFAs from TAGs were positively associated. These findings suggest that fatty acids from different sources and lipid classes have opposite influences on diabetes risk. This would provide novel insight into the role of lipid metabolism in diabetes onset and further develop guidelines for a healthy diet to prevent diabetes.

In addition to investigating the pathology of diabetes onset, we also developed an 11-lipid panel to predict future diabetes. Traditional clinical parameters such as FPG and 2hPG can achieve a prediction power AUC of 0.775. However, when we combine our lipid panel with FPG and 2hPG, we can improve the prediction power from 0.775 to 0.842. Among those 11 lipids, 10 belong to TAG and one is PC. These results suggest that specific metabolites of the TAG and PC classes play important roles in the early detection of women who will transition from GDM to T2D. Since diabetes is a metabolic disorder involving dysmetabolism of carbohydrate, lipids and amino acids, it is not surprising that biomarkers of both carbohydrate and lipid metabolism can improve the predictive power over carbohydrate metabolism alone. Based on our data, we would envision that adding a specific lipidomic signature to existing clinical parameters for testing, perhaps including other metabolites (ie. biogenic amines and amino acids) will provide a more accurate assessment of future T2D risk. Nonetheless, our study provides an important clinical application for early prediction of diabetes when most GDM women return to normoglycemia after delivery. The early prediction will contribute to early intervention and prevention of diabetes.

## Materials and methods

### SWIFT cohort

The Study of Women, Infant Feeding, and Type 2 Diabetes Mellitus After GDM Pregnancy (SWIFT) is a prospective cohort that conducted in-person research exams among 1035 women with GDM diagnosed based on the 3 hr 100 g OGTT via Carpenter and Coustan's criteria, and no prior history of diabetes or other serious health conditions (age 20–45 years, diverse ethnicities) within the Kaiser Permanente Northern California Healthcare System (KPNC) (*Carpenter and Coustan, 1982*). Details of the cohort recruitment, selection criteria, methodologies have been described previously (*Gunderson et al., 2011*). Of 1035 women with GDM who consented to participate in the three in-person research exams for the SWIFT Study, 1010 participants did not have T2D at baseline (6–9 weeks postpartum) based on 2 hr 75 g oral glucose tolerance tests (OGTTs). All research participants underwent annual research 2 hr 75 g OGTTs and other assessments at baseline throughout 2 years of follow-up, and subsequently for medical diagnoses of diabetes confirmed by laboratory testing from electronic medical records up to 8 years post-baseline. Research methodology included monthly quantitative assessment of lactation intensity and duration, socio-demographics, medical conditions, medication use, reproductive history, depression, subsequent births, lifestyle behaviors, body composition and anthropometry (*Gunderson et al., 2011*). Fasting and 2 hr postload plasma samples from 75 g OGTTs (baseline, 1 year, and 2 years post-baseline) were analyzed within several weeks for glucose and insulin levels, and fasting stored samples from the SWIFT Biobank (−80℃) were used to measure a lipid panel, free fatty acids and adipokines, as previously described (*Gunderson et al., 2014*; *Gunderson et al., 2012*). Follow-up assessments to determine new onset T2D status were based on research 2 hr 75 g OGTTs and KPNC electronic medical records data based on mediation, ICD codes and laboratory tests for glucose tolerance (*Gunderson et al., 2015*). T2D diagnosis was based on the American Diabetes Association (ADA) criteria (*Expert Committee on the Diagnosis and Classification of Diabetes Mellitus, 2003*). The study design and all procedures were approved by the Kaiser Permanente Northern California Institutional Review Board (protocol numbers #CN-04EGund03-H and #1279812–10) and Office of Research Ethics at University of Toronto (protocol number #38188). All participants gave written informed consent before taking part in the research exams.

## Lipidomics assay

Baseline fasting plasma from 350 samples from a subset of the cohort (171 incident T2D vs 179 non-T2D controls) were sent to Metabolon, Inc (Morrisville, NC) and measured by GC-MS and LC-MS. Lipids were extracted from the bio-fluid in the presence of deuterated internal standards using an automated BUME extraction according to the method of *Löfgren et al., 2012*. The extracts were dried under nitrogen and reconstituted in ammonium acetate dichloromethane: methanol. The extracts were transferred to vials for infusion-MS analysis, performed on a Shimadzu LC with nano PEEK tubing and the Sciex SelexIon-5500 QTRAP. The samples were analyzed via both positive and negative mode electrospray. The 5500 QTRAP was operated in MRM mode with a total of more than 1,100 MRMs. Individual lipid species were quantified by taking the ratio of the signal intensity of each target compound to that of its assigned internal standard, then multiplying by the concentration of internal standard added to the sample. Lipid class concentrations were calculated from the sum of all molecular species within a class, and fatty acid compositions were determined by calculating the proportion of each class comprised by individual fatty acids. In this study, a total of 1008 lipid species from 15 classes and 296 fatty acids were measured. In particular, in the natural lipid group, 26 cholesterol esters (CE), 26 monoacylglycerol (MAG), 59 diacylglycerol (DAG), 493 triacylglycerol (TAG), and 26 free fatty acids (FFA) were detected. In phospholipid group, 140 phosphatidylcholine (PC), 216 phosphatidylethanolamine (PE), 28 phosphatidylinositol (PI), 26 lysophosphatidylcholine (LPC), and 26 lysophosphatidylethanolamine (LPE) were measured. In sphingolipid group, levels of 13 dihydroceramide (DCER), 12 ceramide (CER), 12 hexosylceramide (HCER), 12 lactosylceramide (LCER), and 12 species of sphingomyelin (SM) were tested.

## Data analyses

Data processing was performed for further statistical analysis. Lipids with >5% missing values were removed from the data allowing only the most robust lipids for the following statistical analysis. After this filtering step, 1008 species were reduced to 816 for further analysis. Remaining missing values were imputed as 1/2 minimum value for each specific lipid. Sample normalization was performed by normalizing each value within the sample to the total value of the sample to adjust differences among the samples. Log-transformation was performed. Odds ratios (ORs) of each lipid for T2D incidence were calculated by applying logistic regression models adjusting effects from race/ethnicity, age and BMI. FDR was calculated by correcting p-value by Benjamini-Hochberg method for multiple comparison. A cut-off of FDR < 0.05 was used for significance. Lipids with FDR of odds ratio <0.001 were subjected for lipid predictor selection. By applying a conditional logistic regression model with stepwise method (including forward and backwards), 11 lipids were selected for prediction models. Classification models were built with logistic regression and cross validation was performed to evaluate the prediction performance. Prediction performance was presented as receiver operating characteristic (ROC) curves. Because association of lipids with diabetes risk can differ based on acyl chain length and unsaturation degree, lipids were grouped and further analyzed based on carbon atom and double bond numbers. All the analyses above were performed in open-source, statistical software, R v3.2.4. Pathway analysis was performed using positive- or negative-associated lipids in the web tool MetaboAnalyst 4.0 (*Chong et al., 2018*).

## Acknowledgements

The authors would like to thank Gabriele V Ronnett for important contributions to data interpretation and Ying Liu for contributions to study conceptualization. The authors thank the participants of the SWIFT Study for their commitment and important contributions to the study.

## Additional information

**Competing interests**

Erica P Gunderson: Erica P Gunderson is affiliated with Kaiser Permanente. EPG has declared a research grant from Janssen Pharmaceuticals Company. EPG has no other financial interests to

declare. Michael B Wheeler: MBW has declared a research grant from Janssen Pharmaceuticals Company. The other authors declare that no competing interests exist.

## Funding

| Funder | Grant reference number | Author |
|---|---|---|
| Canadian Institutes of Health Research | FRN 143219 | Michael B Wheeler |
| Eunice Kennedy Shriver National Institute of Child Health and Human Development | R01 HD050625 | Erica P Gunderson |
| NIDDK | R01 DK118409 | Erica P Gunderson |
| Janssen Pharmaceuticals | 430086739 | Erica P Gunderson Michael B Wheeler |
| Banting and Best Diabetes Centre, University of Toronto | Postdoctoral Fellowship | Mi Lai |
| Ontario Graduate Scholarship | Graduate Student Fellowship | Dana Al Rijjal |
| Banting and Best Diabetes Centre, University of Toronto | Graduate Student Fellowship | Dana Al Rijjal |

The funders had no role in study design, data collection and interpretation, or the decision to submit the work for publication.

## Author contributions

Mi Lai, Conceptualization, Data curation, Software, Formal analysis, Validation, Investigation, Visualization, Methodology, Writing - original draft, Writing - review and editing; Dana Al Rijjal, Validation, Writing - review and editing; Hannes L Röst, Software, Supervision, Methodology, Writing - review and editing; Feihan F Dai, Conceptualization, Supervision, Validation, Writing - original draft, Project administration, Writing - review and editing; Erica P Gunderson, Conceptualization, Resources, Data curation, Supervision, Funding acquisition, Validation, Investigation, Methodology, Writing - review and editing; Michael B Wheeler, Conceptualization, Resources, Supervision, Funding acquisition, Writing - review and editing

## Author ORCIDs

Michael B Wheeler (iD) https://orcid.org/0000-0002-7480-7267

## Ethics

Clinical trial registration ClinicalTrials.gov Identifier: NCT01967030.
Human subjects: The study design and all procedures were approved by the Kaiser Permanente Northern California Institutional Review Board (protocol numbers #CN-04EGund-03-H and #1279812-10) and Office of Research Ethics at University of Toronto (protocol number #38188). All participants gave written informed consent before taking part in the research exams.

## Decision letter and Author response

Decision letter https://doi.org/10.7554/eLife.59153.sa1
Author response https://doi.org/10.7554/eLife.59153.sa2

# Additional files

## Supplementary files

• Transparent reporting form

## Data availability

Lipidomic data have been deposited in Harvard Dataverse: https://doi.org/10.7910/DVN/KUDDSF. Source data files have been provided for Figures 2, 3, 4 and 6 as supporting files.

The following dataset was generated:

| Author(s) | Year | Dataset title | Dataset URL | Database and Identifier |
|---|---|---|---|---|
| Wheeler MB | 2020 | SWIFT lipidomics data | https://doi.org/10.7910/DVN/KUDDSF | Harvard Dataverse, 10.7910/DVN/KUDDSF |

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
