## [Decision Letter]

**Acceptance summary:**

The authors use a quantitative lipidomics assay to identify metabolic differences in plasma that predict whether women with gestational diabetes will progress to frank diabetes in the years after pregnancy. They determine that combining key aspects of the plasma lipid signature with conventional measures of metabolism and glucose homeostasis improve the ability to identify women with the highest risk of subsequent diabetes.

**Decision letter after peer review:**

Thank you for submitting your article "Underlying dyslipidemia postpartum in women with a recent GDM pregnancy who develop Type 2 diabetes" for consideration by *eLife*. Your article has been reviewed by three peer reviewers, including Ralph DeBerardinis as the Reviewing Editor and Reviewer #1, and the evaluation has been overseen by Anna Akhmanova as the Senior Editor.

The reviewers have discussed the reviews with one another and the Reviewing Editor has drafted this decision to help you prepare a revised submission. Overall, the reviewers were enthusiastic about the paper but pointed out several areas where it could be improved for clarity or content.

Summary:

Lai et al. used plasma lipidomics to identify metabolic signatures that predict whether patients with gestational diabetes mellitus (GDM) will progress to Type 2 diabetes (T2DM) in the years after pregnancy. From over 1,000 patients with GDM, the authors performed lipidomics in 171 patients who developed T2DM at follow up and 179 controls who did not. From plasma acquired soon after pregnancy, they identified numerous lipids correlating positively or negatively with T2DM. These included a large number of positively-associated triglycerides and several negatively-associated phosphatidylcholines and sphingomyelins. From the most informative lipids, the authors derived an 11-lipid signature which, when combined with conventional markers of diabetes in the post-pregnancy state, improved the ability to predict future diabetes. The authors conclude that patients destined to progress from GDM to T2DM have aberrant plasma lipid profiles soon after pregnancy, and that detecting these abnormalities can be used to predict future T2DM.

Essential revisions:

1) Regarding the choice of cohorts, is unclear how the authors matched 171 women with incident T2D to 179 controls. The number of subjects are unequal. The construction of the dataset should be better described.

2) There needs to be a more informative way to communicate how long the patients were followed after pregnancy than "up to 8 years." Perhaps the authors can indicate from the cohort analyzed by lipidomics how many patients were followed for 1, 2, 3 etc years, because this could influence the interpretation of the incidence of T2DM. The authors should also confirm that the total years of follow-up were similar between the cases and controls in this cohort.

3) There also needs to be more detail about the lipidomic method, beyond that it was performed by a company. For example, are any of these lipids subjected to formal quantitation or are these all relative values? How was normalization performed? If the abundance of each species was normalized to a total lipid content, then the apparent reduction of some phospholipids may be an artifact of the elevated TAGs.

4) Furthermore, although the authors outlined in the Introduction that there have been a number of different lipidomics studies in diabetes, it is less clear whether there was convergence on a set of lipids among these different studies. Can you summarize? If every study lands on different lipids, that could indicate that 1) the underlying biology is different among cohorts with different inherent risks; 2) technical differences in the mass spectrometry; or 3) study design. Some discussion about these issues would improve the paper.

5) Figure 5 should include the glucose pathway. From the data, the obvious implication is that higher glucose levels lead to higher G3P levels which shift palmitoyl-CoA (and other fatty acid CoAs) to TAGs and away from sphingolipids. Also if higher glucose levels correlate with higher TAG levels, are TAGs simply an indirect measure of glucose or Is the connection more complicated?

6) The authors state that "samples were randomly divided into 80% as training set for model building and 20% as hold-out testing set for prediction validation. " This is an accepted approach to assess the performance of a prediction model. However, with a single divide, the conclusion might be unduly affected by random variability. The cross-validation approach can be used to better assess the prediction performance across random splittings.

7) It is important that the authors provide some context about the value of the increased predictive performance that comes with the lipid signature. In other words, do the data support applying a broad lipidomic method like the Metabolon platform, a focused lipid signature, or some other combination of predictive markers to assess risk, given that the conventional FPG and 2hPG seem to do a good job of prediction already?

8) Although definitive mechanistic explanations for these associations are beyond the scope of the current paper, some speculation would help the Discussion.

---

## [Author Response]

Essential revisions:1) Regarding the choice of cohorts, is unclear how the authors matched 171 women with incident T2D to 179 controls. The number of subjects are unequal. The construction of the dataset should be better described.

Our rationale was to maximize the statistical power in our analysis and as such included 8 additional control samples. We have now performed the analysis with 171 incident T2D vs. 171 controls and found that the significant lipids are the same as that we obtained in the original analysis (with 171 incident T2D vs. 179 controls). We agree with the reviewer/editor, even though we tried to pair match case and control groups on age, race and BMI, it is not a strict 1 vs. 1 perfect match. We have revised the wording (change “pair match” to “versus”) in the manuscript to be more precise.

2) There needs to be a more informative way to communicate how long the patients were followed after pregnancy than "up to 8 years." Perhaps the authors can indicate from the cohort analyzed by lipidomics how many patients were followed for 1, 2, 3 etc years, because this could influence the interpretation of the incidence of T2DM. The authors should also confirm that the total years of follow-up were similar between the cases and controls in this cohort.

We apologize for not communicating some of the details of the SWIFT analysis clearly. For both case and control groups, the total years of follow-up were similar. All research participants underwent 2-h 75-g OGTTs and other assessments at baseline and thereafter annually for 2 years and subsequent medical diagnoses of diabetes was retrieved from electronic medical records for 8 years post-baseline. Specifically, all participants attended three in-person clinics (2h 75g OGTT and other clinical assessments) at baseline, 1-year and 2 years post-baseline. After 2 years post-baseline, for the cases, subjects were followed until they were diagnosed as T2D, specifically from electronic medical records available. For control subjects, all the patients were followed for 8 years. We added this point in the Results (subsection “Clinical characterization of the participants at baseline”).

3) There also needs to be more detail about the lipidomic method, beyond that it was performed by a company. For example, are any of these lipids subjected to formal quantitation or are these all relative values? How was normalization performed? If the abundance of each species was normalized to a total lipid content, then the apparent reduction of some phospholipids may be an artifact of the elevated TAGs.

This is an important point. We have now added more details with regard to the lipidomics method in the Materials and methods section (subsection “Lipidomics assay”). The assay uses deuterated internal standards and the concentrations of individual lipid species are provided. The units for the lipids concentration are in μM units. Before we subjected the data into analysis, we performed the sample normalization (normalized each value within the sample to the total value of the sample) to allow general-purpose adjustment for differences among the samples. These will not affect the proportion of each lipid in each sample and the statistical analysis between samples (case vs. control group). We didn’t normalize each lipid to the total lipid content and we have carefully revised the Materials and methods part to clarify this.

4) Furthermore, although the authors outlined in the Introduction that there have been a number of different lipidomics studies in diabetes, it is less clear whether there was convergence on a set of lipids among these different studies. Can you summarize? If every study lands on different lipids, that could indicate that 1) the underlying biology is different among cohorts with different inherent risks; 2) technical differences in the mass spectrometry; or 3) study design. Some discussion about these issues would improve the paper.

We appreciate the reviewer’s comments and agree that the message could be clarified here. As suggested, we have now summarized the findings from different lipidomics studies in diabetes in the Introduction section (second paragraph). In all of these studies, a positive association of TAG/DAG and T2D risk was consistently observed. However, a convergence on specific lipids were not evident. This could be due to the differences in study design, cohort background and methodology including, importantly, limitations in coverage – expressed lipids in each study were not consistent.

5) Figure 5 should include the glucose pathway. From the data, the obvious implication is that higher glucose levels lead to higher G3P levels which shift palmitoyl-CoA (and other fatty acid CoAs) to TAGs and away from sphingolipids. Also if higher glucose levels correlate with higher TAG levels, are TAGs simply an indirect measure of glucose or Is the connection more complicated?

We would like to thank the reviewer for these insightful comments. We fully agree with this interpretation and have now revised the figure by adding glucose into the pathway. We have also added pertinent points in the Discussion addressing this view (fifth paragraph). Although glucose levels could correlate with TAG levels, our correlation analysis with coefficient r values revealed that the correlation level was weak. The connection between these two factors is likely more complicated. Increased TAG along with decreased phospholipids and sphingolipids represented a profound lipid dysmetabolism.

6) The authors state that "samples were randomly divided into 80% as training set for model building and 20% as hold-out testing set for prediction validation. " This is an accepted approach to assess the performance of a prediction model. However, with a single divide, the conclusion might be unduly affected by random variability. The cross-validation approach can be used to better assess the prediction performance across random splittings.

We agree, this is an important point. Based on the reviewer’s comment, we have now redone the prediction analysis with cross-validation used to assess the performance. We have revised Figure 6B and related parts in the Results and Materials and methods section (subsections “Selective lipids can predict future diabetes and complement clinical diagnostics” and “Data Analyses”).

7) It is important that the authors provide some context about the value of the increased predictive performance that comes with the lipid signature. In other words, do the data support applying a broad lipidomic method like the Metabolon platform, a focused lipid signature, or some other combination of predictive markers to assess risk, given that the conventional FPG and 2hPG seem to do a good job of prediction already?

These are very insightful comments and important for the translatability of this study with respect to assessing risk of T2D in women with a history of GDM. To provide some context, the lipids show higher AUC than FPG or 2hPG alone. Adding lipids to FPG improved the AUC from 0.703 to 0.795. Adding lipids to 2hPG improved the AUC from 0.704 to 0.809. This clearly show that the lipids significantly increase the predictive performance of the classic measurements. We agree with the reviewer, the combination of FPG and 2hPG can achieve prediction power AUC 0.775, the adding of lipids did increase the performance to AUC0.842 (Figure 6—source data 1). Based on our data, we would envision that adding a specific lipidomic signature to existing clinical parameters for testing, perhaps including other metabolites (i.e. biogenic amines and amino acids) will provide a more accurate assessment of future T2D risk. We have now discussed in the Discussion section (last paragraph).

8) Although definitive mechanistic explanations for these associations are beyond the scope of the current paper, some speculation would help the Discussion.

Thanks for the comment. We have revised the Discussion section as the reviewer suggested. Specifically, we speculated the potential mechanism of increased serum TAG level through lipogenesis activated by high serum sugar level (Discussion, second paragraph). We have also discussed the mechanism of the pathway alteration in glucose, TAG, phospholipids and sphingolipids as the reviewer suggested. In short, we speculated that phospholipids (through DAG) and TAGs shared the same precursor glycerol-3-phosphate (G3P) and the higher G3P induced by higher plasma glucose levels could shift the Fatty acid-CoA to lipogenesis from sphingolipids and phospholipids pathways (Discussion, fifth paragraph).